# Variability of manometric sea level from reanalyses and observation-based products over the Arctic and North Atlantic Oceans and the Mediterranean Sea

Andrea Storto[1], Giulia Chierici[1], Julia Pfeffer[2], Anne Barnoud[2], Romain Bourdalle-Badie[3], Alejandro Blazquez[4], Davide Cavaliere[1], Noémie Lalau[2], Benjamin Coupry[2], Marie Drevillon[3], Sebastien Fourest[4], Gilles Larnicol[2], Chunxue Yang[1]

[1] Institute of Marine Sciences (ISMAR), National Research Council (CNR), Rome, Italy

[2] Magellium, 31520 Ramonville-Saint-Agne, France

[3] Mercator Ocean International (MOI), 31400 Toulouse, France

[4] Laboratory of Space Geophysical and Oceanographic Studies (LEGOS), 31401 Toulouse, France

*Correspondence to*: Andrea Storto, Institute of Marine Sciences (ISMAR), National Research Council (CNR), via del Fosso del Cavaliere 100, I-00133 Roma, Italy; Email: andrea.storto@cnr.it

**Abstract.** Regional variations of the mass component of sea level (manometric sea level) are intimately linked with the changes in the water cycle, volume transports, and inter-basin exchanges. Here, we investigate the consistency at the regional level of the manometric sea level from the Copernicus Marine Service Global Reanalysis Ensemble Product (GREP) and compare with observation-based products, deduced from either gravimetry (GRACE missions) or altimetry and in-situ ocean observations (sea level budget approach, SLB), for some climate-relevant diagnostics such as interannual variability, trends, and seasonal amplitude. The analysis is performed for three basins (Mediterranean Sea; Arctic, and North Atlantic Oceans), and indicates very different characteristics across the three. The Mediterranean Sea exhibits the largest interannual variability, the Arctic Ocean the largest trends, and the North Atlantic a nearly linear increase that is highly correlated to global barystatic sea level variations. The three datasets show significant consistency at both the seasonal and the interannual time scales, although differences in linear trends are sometimes significant (e.g., GRACE overestimates the trend in the Arctic and underestimates it in the Mediterranean Sea, compared to the other products). Furthermore, GRACE and GREP prove mutually more consistent than SLB in most cases. Finally, we analyze the main modes of climate variability affecting the manometric sea level variations over the selected ocean basins through regularized regression; the North Pacific Gyre Oscillation, the Arctic Oscillation, and the Atlantic Multidecadal Oscillation are proven to be the most influential modes for the North Atlantic Ocean, Mediterranean Sea, and Arctic Ocean manometric sea level, respectively.

**Short summary.** The variability of the manometric sea level (i.e., the sea level mass component) in three ocean basins is investigated in this study using three different methods (reanalyses, gravimetry, and altimetry in combination with in-situ observations). We identify the emerging long-term signals, the consistency of the datasets, and the influence of large-scale climate modes on the regional manometric sea level variations at both seasonal and interannual time scales.

# 1 Introduction

Contemporary changes in global sea level are driven mostly by two contributions: density-driven variations of sea level, the so-called steric sea level that responds to the expansion and contraction of seawater due, mostly, to increasing heat in the oceans (Storto et al., 2019a). The other contributor to global sea level change is the ocean mass change, called barystatic sea level (Gregory et al., 2019). Barystatic sea level has been recently found to be responsible for the majority (about 60%) of the global sea level changes (Frederikse et al., 2020; Fox-Kemper et al., 2021). Recent estimates indicate $2.25 \pm 0.16$ mm yr$^{-1}$ of sea level rise due to barystatic changes for the recent period (2005-2016; Amin et al., 2020). Changes in barystatic sea level are due to the loss of mass from glaciers and ice sheets (Greenland and Antarctica) and changes in the global water cycle and land water storage. As such, barystatic sea level changes are a fundamental proxy of climate change and are expected to increase even more dramatically in the future due to increased ice melting, according to future projections (Oppenheimer et al., 2019).

At the regional scale, local dynamics, and regional hydrology, together with cross-basin exchanges, modulate regional ocean mass exchanges, called manometric sea level (Gregory et al., 2019). For instance, Camargo et al. (2022) show that regional trends in manometric sea level may vary from -0.4 to 3.3 mm yr$^{-1}$ across the global ocean for the 2003-2016 period. Typically, regions characterized by high dynamic variability are characterized by large manometric variations. Strong climate modes of variability (e.g., the North Atlantic Oscillation) are also responsible for large deviations in manometric sea level (e.g., Criado-Aldeanueva et al., 2014; Volkov et al., 2019); fingerprinting techniques can be used to estimate the influence of a specific climate index on the resulting sea level variability (e.g., Pfeffer et al., 2022). In the Mediterranean Sea, for instance, variations are intimately linked to the exchanges with the Atlantic Ocean through the Gibraltar Strait, and variations in the atmospheric freshwater input, which are both strongly linked to the North Atlantic variability (e.g., Tsimplis and Josey, 2001).

Since 2002, methods to observe and analyze manometric and barystatic sea level variations have generally relied on GRACE (Gravity Recovery And Climate Experiment; e.g. Tapley et al., 2004) and GRACE-FO (GRACE-Follow On; Landerer et al., 2020) satellite mission measurements of the temporal and spatial variations of the Earth's gravity field. Barystatic and manometric sea level signals can also be inferred from the difference between total sea level, measured by altimetry missions, and steric sea level, estimated through in-situ observations (e.g., Horwath et al., 2022). This approach will be referred to as the Sea Level Budget (SLB) method in the remainder of this article.

Alternatively, ocean general circulation model (OGCM) simulations embed the variability of sea level and its components, although they significantly lack realism (e.g., Kohl et al., 2007). Ocean reanalyses, which combine an ocean model with observations through data assimilation (Storto et al., 2019b) are in turn able to provide a good estimation of ocean long-term changes (e.g., Storto and Yang, 2024) and associated sea level variability at global and basin scales (e.g., Storto et al., 2017); they are thus complementary to gravimetry and sea level budget-based observational counterparts and can be used for several investigations (e.g., Peralta-Ferriz et al., 2014; Marcos, 2015; Hughes et al., 2018). A few limitations in the use of reanalyses exist, though. First, the usual Boussinesq approximation in the OGCMs leads to a zero global steric sea level by construction, as the models cannot represent the global expansion and contraction in the constant volume framework. However, the global steric sea level can be computed and added to the model sea surface height retrospectively, since it does not have any dynamical signature (e.g., Greatbatch, 1994).

A more critical and long-standing issue in reanalyses regards the barystatic and manometric sea level components. Indeed, both the use of climatological freshwater input from land and ice and the imbalance of the atmospheric freshwater forcing combined with the evaporation and sublimation calculated by the ocean model make barystatic and manometric terms often unrealistic. Some reanalyses correct the barystatic sea level with globally uniform offsets, either time-varying or constant. In any case, the barystatic signal is generally unrealistic, and the manometric one may be affected by inaccuracies in the freshwater input into the oceans. In general, ocean bottom pressure data derived from gravimetry could also be directly

assimilated into ocean models (see e.g., Köhl et al., 2012). However, this approach was found suboptimal, mostly due to the low signal-to-noise ratio of the gravimetry data compared to altimetry data assimilation (e.g., Storto et al., 2011), and their issues related to the pre-processing (persistent stripes and land water leakage). More recently, however, ingesting gravimetry data (e.g., in ECCOv4r4, ECCO Consortium, 2020) has proven promising to better capture high-frequency sea level variability (Schindelegger et al., 2021). Finally, the limited spatial resolution of the models may limit the representativeness of sea level variations in mesoscale active areas (e.g., Androsov et al., 2020).

The goal of this paper is manifold. First, we aim to estimate the consistency of manometric sea level from notably different approaches, which use numerical ocean models, gravimetry or altimetry, and in-situ observations. These approaches are known to contain different sources of uncertainty and none of them is fully trustable, as discussed in detail in this and the next sections. Particular attention is devoted to assessing whether the latest generation of the Copernicus Marine Service global reanalyses can capture the interannual variations of the manometric sea level. Second, we aim at quantifying regional trends and amplitudes, to identify the emerging levels and scales of manometric sea level variability depending on the specific basin. Finally, we aim to fingerprint the manometric sea level with several climate mode indices, to connect such variations with large-scale climate variability.

The structure of the paper is as follows: we compare regionally (section 3) the manometric sea level from reanalyses with those coming from satellite gravimetry or the sea level budget approach (described in section 2). The exercise will therefore indicate the consistency of the reanalyses and observation-based products for selected metrics. Finally, we summarize and conclude (section 4).

## 2 Data and Methods

In this section, we shortly introduce the datasets used in the assessment. We refer to Gregory et al. (2019) for the terminology and definitions used to characterize the sea level components.

### 2.1 Gravimetry-based dataset

Barystatic and manometric sea level anomalies have been estimated from April 2002 to August 2022 at a monthly timescale and with a spatial resolution of 1° using an ensemble of GRACE and GRACE-FO solutions (Product ref. no. 2 in Table 1). The GRACE and GRACE-FO ensemble is constituted of 120 solutions, allowing us to estimate the uncertainties associated with different processing strategies and geophysical corrections needed for ocean applications. The ensemble is based on coefficients of the Earth's gravitational potential anomalies estimated by five different processing centers (CNES, CSR, JPL, GFZ, ITSG). A large variety of post-processing corrections are applied to the ensemble, including two geocenter motions (Lemoine and Reinquin, 2017; Sun et al., 2016), three oblateness values (C20) of the Earth (Cheng et al., 2013; Lemoine and Reinquin, 2017; Loomis et al., 2019), and two Glacial Isostatic Adjustment (GIA) corrections (Peltier et al., 2015, Caron et al., 2018). To reduce the anisotropic noise, characterized by typical stripes elongated in the North-South direction, decorrelation filters, called DDK filters (Kusche et al., 2009), are applied to GRACE solutions (e.g., Horvath et al., 2018), using two different orders (DDK3 and DDK6) corresponding to different levels of filtering. The ensemble of 120 solutions results from the combination of these five processing centers, two geocenter models, three oblateness models, two GIA corrections, and two filters. The ensemble standard deviation provides a measure of uncertainty for both the barystatic and manometric sea level timeseries.

### 2.2 Sea level budget-based dataset

The estimation of barystatic and manometric sea level changes is extended to the altimetry era (January 1993 - December 2020) using the sea level budget approach (Product ref. no. 3 in Table 1). The manometric sea level changes are calculated as the difference between the geocentric sea level changes based on satellite altimetry and steric sea level changes based on in-situ measurements of the seawater temperature and salinity. The reliability of this dataset is intrinsically linked to the altimetry and in-situ observational sampling. Only within the global mean values, i.e. the barystatic sea level, changes are computed as the difference between the global mean geocentric sea level changes and thermosteric sea level changes to avoid drifts due to Argo salinity measurement errors (Barnoud et al., 2021; Wong et al., 2020); however, regional (manometric) sea level estimates include the halosteric contribution in the steric evaluation.

Geocentric sea level changes are estimated using the vDT2021 sea level product provided by the Copernicus Climate Change Service (C3S; Legeais et al., 2021). Geocentric sea level changes are corrected for the drifts in Topex-A altimeter (Ablain, 2017) and Jason-3 microwave radiometer wet tropospheric correction (Barnoud et al., 2023a, 2023b), for the GIA effect, using the ensemble mean of 27 GIA models (Prandi et al., 2021) centered on ICE5G-VM2 (Peltier et al., 2004), and for the elastic deformation of the solid Earth due to present-day ice melting (Frederikse et al., 2017). The uncertainty of the geocentric sea level changes is calculated with the uncertainty budget and method detailed in Guérou et al. (2023) for the global mean sea level changes and in Prandi et al. (2021) for the local sea level changes. Altimetry data is masked over sea-ice-covered areas using the Copernicus Climate Change Service sea-ice product (Lavergne et al., 2019).

Steric sea level changes are estimated as the sum of the thermosteric and halosteric sea level changes calculated from gridded temperature and salinity estimates from three different centers including EN4 (Good et al., 2013), IAP (Cheng et al., 2020) and Ishii et al. (2006). EN4 provides four datasets with different combinations of corrections for XBT and MBT measurements applied, leading to an ensemble of 6 temperature and salinity datasets. From these datasets, we compute the thermosteric and halosteric sea level changes due to temperature and salinity variations between 0 and 2000 m depth. The deep ocean contribution (i.e., below 2000 m) is considered only in the global barystatic signal and taken as a linear trend of $0.12 \pm 0.03$ mm yr$^{-1}$ (Chang et al., 2019) added to the time-varying steric sea level; for the regional estimates of the manometric sea level, the deep and abyssal ocean contribution is neglected, as there are no enough data for constraining it at regional level.

Steric sea level changes are estimated as the ensemble mean of the 6 solutions, and their uncertainties are estimated with the covariance matrix of the ensemble. The resulting barystatic and manometric uncertainties are described by the covariance matrix obtained by summing the sea level and steric covariance matrices; the sea-ice mask from the altimetry product is propagated onto the resulting manometric product.

**2.3 The reanalysis dataset**

In this work, we use the Global Reanalysis Ensemble Product (GREP) from the Copernicus Marine Service (Product ref. no. 1 in Table 1), which is a small-ensemble global reanalysis product, including in turn the four reanalyses i) C-GLORS (v7) from CMCC; ii) GloSea5 from UKMO; iii) GLORYS2 (V4) from Mercator Ocean, and iv) ORAS5 from ECMWF. All reanalyses are performed using the NEMO ocean model (Madec et al., 2017) configured at about 1/4° of horizontal resolution and 75 levels. However, the four reanalyses differ for several issues, which can be summarized in the i) NEMO model version and a few selected parametrizations, including specific choice in the use of the ECMWF reanalysis (ERA-Interim and ERA5) atmospheric forcing; ii) initial conditions in 1993 at the beginning of the reanalyzed period (1993-2019); iii) the data assimilation scheme, and iv) the set of observations assimilated, including their source and pre-processing procedures. Thus, GREP can span, to a good extent, the uncertainty linked with model physics and input datasets. We have used monthly mean data at 1/4° of horizontal resolution for the comparison described in the following section. More details about the four reanalyses, together with some in-situ-based validation and assessment of the ensemble standard deviation, are provided by Storto et al. (2019c).

The estimation approach for GREP follows that of the sea level budget approach (see section 2.2), where the manometric sea level is calculated as a difference from the total sea surface height anomaly from the reanalysis, and the steric sea level anomaly, calculated from the reanalysis output temperature and salinity fields. Thus, we can cross-compare GREP data with GRACE and SLB datasets in terms of interannual variability, trend, and seasonal amplitude.

## 2.4 Analysis methods

Basin-averaged timeseries are analyzed in the next section as monthly means to assess the main variability signal over three oceanic basins (the Arctic Ocean, defined as the region covering from 67°N in the Atlantic to the Bering Strait; the North Atlantic Ocean, defined from 0°N to 67°N; the Mediterranean Sea). Timeseries are also analyzed in terms of their interannual and seasonal signal, where the interannual signal is the timeseries to which the monthly climatology has been subtracted and the seasonal is the residual part, assuming that the majority of the subannual signal can be attributed to seasonal variation, due to the monthly temporal frequency of the data. The uncertainty of the timeseries corresponds to that provided by the dataset (which in turn uses an ensemble approach to estimate uncertainty as ensemble standard deviation); by construction, GREP, with only four members, is known to underestimate the uncertainty of sea level (Storto et al., 2019c). Uncertainty of trends is estimated through bootstrapping (Efron, 1979) and closely resembles the estimates calculated following Storto et al. (2022). The bootstrapping technique randomly removes part of the timeseries, and thus quantifies the sensitivity of the trend to individual years and periods. Explained variance is used to quantify how much of the regional signal is explained by the global barystatic signal due to fast barotropic motion. For this analysis, we use only global GRACE and SLB timeseries and show only SLB for the sake of clarity (see, e.g. Barnoud et al., 2023b, for a discussion on their comparison), because the GREP barystatic sea level is either unreliable due to drifts in the freshwater forcing, or it is adjusted to GRACE-derived data and, thus, is not independent. Seasonal amplitude is defined by fitting the monthly data to a curve with sinusoidal (seasonal signal) and linear (trend signal) terms; the interannual variability is the standard deviation of the detrended and deseasonalized timeseries. Percents of manometric sea level trends over the total sea level ones are calculated from the Copernicus Marine Service dataset (Product ref. no. 4 in Table 1), over each region.

LASSO regression (Tibshirani, 1997), performed between the normalized manometric sea level and normalized climate indices, is a regularization technique for multivariate regression, which is used in this study to rank the influence of the climate indices on the basin-averaged manometric sea level, in a way similar to what Pfeffer et al. (2022) proposed. Like the latter and previous studies, raw monthly means were used without low-pass filtering the data, which could induce arbitrary preferences in the regression within our multi-variate analysis. After performing k-fold cross-validation (with 10 folds) to identify the best hyperparameters, LASSO regression avoids overfitting the regression, such that absolute values of the regression coefficients quantify the impact of a predictor on the manometric sea level. By construction, LASSO minimizes the collinearities across the predictors; however, when predictors are strongly correlated, the preference provided by LASSO might be less obvious than expected (Tibshirani, 1996). We also verified that other methods (e.g., the $R^2$ hierarchical decomposition from Chevan and Sutherland, 1991) provide the same results. For these analyses, the *glmnet* (Friedman et al., 2010) and *relaimpo* (Groemping, 2006) R packages are used. Finally, for the statistical significance of the correlations and their differences, we used the *psych* R package (Revelle, 2023) that implements Steiger's test for comparing dependent correlations (Steiger, 1980; Olkin and Finn, 1995). All statistical significance results are provided at the 99% confidence level. The time series and spatial patterns of the climate modes are as in Pfeffer et al. (2022) (see Figures 1, 3, and 4 therein).

## 3 Results

We present the results of the assessment, by first analyzing the timeseries and several diagnostics of the basin-averaged manometric sea level. Then, the consistency between the manometric sea level products is addressed; finally, the influence of

the climate modes of variability on the manometric sea level variability is analyzed. All results presented refer to the 2003-2019 period, common to the three datasets.

## 3.1 Manometric sea level timeseries

The monthly means of the manometric sea level for the three basins considered in this study is shown in Figure 1, while several diagnostics (trend, seasonal amplitude, interannual variability, and mean uncertainty, i.e. the time-avearged uncertainty, estimated in turn as ensemble standard deviation for each product according to Section 2) are provided in Table 2, for the three datasets considered.

The three basins (Arctic Ocean, North Atlantic Ocean, and Mediterranean Sea) exhibit different behavior; GRACE, SLB, and GREP show, however, qualitatively good consistency in all three seas. The Arctic Ocean has a regular periodicity and a large seasonal amplitude, with a generally increasing yearly mean signal, except during the first years of the timeseries (2003-2005). For both GRACE and GREP, the latest years are the ones with the largest manometric sea level, which is reflected in large trends ($3.45 \pm 0.57$ and $2.45 \pm 0.44$ mm yr$^{-1}$, respectively) compared to the other seas, while SLB shows a weaker trend. Manometric sea level changes at interannual time scales are very different over the Arctic Ocean than the global ocean (Table 3), meaning that internal dynamics, straits connections, and the sea-ice seasonal cycle significantly modulate the regional manometric sea level. Seasonal timeseries are more largely explained by the global signal for both datasets (38-48%).

The North Atlantic manometric sea level signal has a seasonality (10 to 14 mm, depending on the dataset), smaller than the other basins, the smallest interannual variability (6.0 to 6.6 mm), and a nearly linearly increasing mean signal that dominates the variability. The percent variance explained by the global barystatic sea level is large (71% and 79% for GRACE and SLB, respectively, for the interannual signal), meaning that the North Atlantic largely resembles the global signal. Here, the manometric trend accounts for about 60-80% of the total sea level trend (provided by altimetry), depending on the specific product used.

In the Mediterranean Sea, the interannual variability is the largest (more than 20 mm for all datasets) and does not follow the global barystatic signal (see the low percent explained variance in Table 3), especially for the interannual signal, no matter which dataset is considered. This suggests that the regional water cycle and sea level budget are mostly independent of the global one, and this is ascribed to the role of Gibraltar Strait (see e.g., Landerer and Volkov, 2013). Trends in the Mediterranean Sea are generally lower than in the other basins and explain about 40%, on average, of the total sea level trend from altimetry. All the datasets exhibit the largest trends in the western part of the Mediterranean Sea (not shown), although with slightly different patterns. Remarkable peaks of the manometric sea level are visible in 2006, 2010, 2011, and 2018; for these events, GREP tends to underestimate the maxima compared to the other two datasets, likely due to the use of climatological discharge from rivers in the reanalyses, and the low resolution at Gibraltar Strait affecting the representation of the Mediterranean inflow. In terms of the uncertainty (see Table 2 and Figure 1), the GRACE dataset exhibits the largest mean uncertainty (about 30 mm in all basins), while the uncertainty of SLB ranges from about 12 mm in the Arctic Ocean and the Mediterranean Sea to about 21 mm in the North Atlantic Ocean. GREP uncertainty is the lowest, except in the Mediterranean Sea where it is comparable to SLB. However, the uncertainty estimates are strongly affected by the ensemble size, which is substantially different across the three datasets (see section 2). Besides, common errors, associated for example with spatial under-sampling, which may be large for the SLB method, will be neglected with the ensemble approach.

## 3.2 Consistency between timeseries

The consistency between the three timeseries is investigated by decomposing the full signal into the interannual and seasonal timeseries. The correlation matrix for the three temporal scales and the three basins is shown in Figure 2.

In the North Atlantic Ocean and the Mediterranean Sea, the largest correlations are generally between SLB and GREP. SLB and GREP are not independent due to the use of altimetry and in-situ observations in both, so this result likely reflects their dependency. At the interannual timescale, the correlation between GRACE and SLB is slightly larger (but the difference is not statistically significant) than that between GRACE and GREP, suggesting that for these regions SLB might capture the year-to-year variations better than the reanalyses. At the seasonal scale in the Mediterranean Sea, however, the consistency between GRACE and GREP is larger than that between GRACE and SLB (with a statistically significant difference), suggesting that the reanalyses capture the seasonal cycle better than SLB with respect to gravimetry data. For both regions, the high consistency of manometric sea level from reanalyses compared to the two observation-based datasets suggests the good reliability of the GREP ensemble mean in capturing the sea level variations.

In the Arctic Ocean, a large consistency is found between GRACE and GREP; the correlations involving SLB are statistically significantly lower than the others, at all time scales (full, inter, and seasonal) at the 99% confidence level; this is also visible, in Figure 1, as fluctuations of the SLB timeseries not reproduced by the other two datasets. On the one hand, the meridional transports, sea-ice modeling, and atmospheric forcing, implicit in the reanalysis systems, are known to be able to shape the Arctic Ocean interannual variability realistically (see e.g. Mayer et al., 2016; 2019); on the other hand, altimetry and in-situ data are poorly sampled in the Arctic Ocean, making more challenging to apply the SLB approach therein. By separating the total sea level and the steric sea level contributions for the SLB and GREP methods (not shown), we have found good consistency for the total sea level inter-annual signal (correlation coefficient equal to 0.69) compared to the steric component (0.35); this suggests that the SLB method has problems over the Arctic basin in representing steric sea level variations, possibly due to the poor in-situ observational sampling.

**3.3 Influence of climate indices on manometric sea level variations**

Several climate indices are considered predictors for the manometric sea level in the three basins (Arctic Ocean, North Atlantic Ocean, and Mediterranean Sea). Their acronyms and meanings are listed in the caption of Figure 3. The detailed justification for inclusion in the analysis is provided by Han et al. (2017), Cazenave and Moreira (2022), and Pfeffer et al. (2022), among many others: through representing well-determined atmospheric circulation regimes and internal climate variability, the indices synthesize the water cycle and the atmospheric forcing variability regimes, leading in turn to variations in the regional manometric sea level due to changes in oceanic divergence and freshwater forcing. For instance, the El Niño Southern Oscillation (ENSO) has a prominent role in modifying precipitation patterns, with obvious implications on the manometric sea level (e.g., Muis et al., 2018); changes in the North Atlantic Oscillation (NAO) modify atmospheric and oceanic transports in North America and Europe, implying changes also in the Mediterranean Sea through modification to exchanges at Gibraltar and precipitation patterns (Landerer et al., 2013; Storto et al., 2019a). It is beyond the scope of this study to explain all possible modes of co-variability and the interested readers are referred to the specific literature for a broad overview (e.g., Andrew et al., 2006; Peralta-Ferriz et al., 2014; Merrifield et al., 2018; Volkov et al., 2019; Pfeffer et al., 2022). Raw monthly means of manometric sea level are used in this study, to avoid arbitrary filtering affecting the regression results; the climate indices, however, are used with filtering as in their standard definition.

In the Arctic Ocean, the largest influence is found to be due to the Atlantic Multidecadal Oscillation (AMO), with values ranging from 25 to 35% depending on the dataset. AMO is known to modulate the sea-ice interannual variations and the Arctic amplification (Li et al., 2018; Fang et al., 2022), which are both important contributors to the sea level manometric fluctuations, due to the increased melting of land ice and disturbances in atmospheric and ocean circulation that jointly influence the variability of manometric sea levels (see, e.g., Previdi et al., 2021). IOD, NAO, and NPGO also significantly affect the Arctic manometric sea level, although the consensus between the datasets varies, and the influence of the IOD is questionable. The Arctic Oscillation is found influential when using the GRACE dataset consistently with previous studies (Peralta-Ferriz et al., 2014), although the other datasets show, in general, other preferences.

The North Atlantic manometric sea level is characterized by the largest impact of NPGO, consistently across all the datasets. While NPGO well explains variations in the eastern North Pacific Ocean (Di Lorenzo et al., 2008), its impact on the North Atlantic manometric sea level likely depends on its global and large-scale influence (Iglesias et al., 2018; Litzow et al., 2020; Pfeffer et al., 2022), which in turn drives to large extent the North Atlantic manometric sea level variability (see Table 3). NPGO accounts for more than 25% of the North Atlantic manometric sea level variability, peaking at more than 40% for the SLB dataset. Significant impact in the North Atlantic manometric sea level is also given by variations described by the PDO, AMO, and IOD, although for the latter small consistency is found across the datasets.

Finally, in the Mediterranean Sea, the largest influence is provided by the Arctic Oscillation (AO), which explains more than 30% of the manometric sea level covariations for all datasets. AO is an expression of the North Atlantic variability, strictly linked to the NAO and closely linked to the North European wind circulation (e.g., Ambaum et al., 2001); while these are strictly connected, the regularization technique used here clearly indicates AO as a better predictor than NAO for the regional manometric sea level; however, this might be an artifact of the LASSO minimization that chooses only one among strongly correlated predictors. Other influential climate modes of variability in the Mediterranean Sea are linked to the North Pacific variability, namely the PDO and NPGO, likely due their effect on the North Atlantic variability.

## 4. Summary and Discussion

In this study, we have focused on the basin-averaged manometric sea level for a few regional basins (Arctic Ocean, North Atlantic Ocean, Mediterranean Sea) and from different datasets, to investigate the consistency, the emerging climate signals, the differences between the basin characteristics, and the link with the main large-scale modes of variability. These three basins were chosen as part of the focus of the EU Copernicus Marine Service and are large enough to be resolved at basin scale by the observing and modeling systems used herein, unlike other smaller basins.

To the authors' knowledge, it is the first time that different datasets of manometric sea level from reanalyses, gravimetry, and altimetry minus in-situ data, are compared at the regional level to infer their strengths and weaknesses. The three basins (Arctic Ocean, North Atlantic Ocean, Mediterranean Sea) exhibit inherently different features, with the Mediterranean Sea showing, on average over the three products, the largest interannual variability, and the smallest trends; the Arctic Ocean shows large seasonal amplitude and the largest trend, and the North Atlantic Ocean a quasi-linear trend, which is very well explained by the global barystatic signal. The three products are found in reasonable agreement, with all pairs significantly correlated at both interannual and seasonal time scales. There are, however, non-negligible differences in the quantitative assessment; for instance, GRACE leads to a large trend in the Arctic basin ($3.45 \pm 0.57$ mm yr$^{-1}$), which is not reproduced by either GREP or SLB and needs to be investigated in more detail; or a trend in the Mediterranean Sea smaller than the others.

In the Arctic Ocean, altimetry minus in-situ (SLB) is generally less in agreement with the other datasets based on correlation scores; this seems to be due to the poor in-situ observation sampling, on which the SLB approach is based (see the PUM, Table 1), which could be alleviated in reanalyses, to some extent, by the atmospheric forcing information and the meridional exchanges. In the Mediterranean Sea, seasonal scale agreement is also the largest between GRACE and GREP, suggesting in turn that the Copernicus Marine Service global reanalyses can capture the manometric sea level variability in the studied regions.

Finally, a fingerprinting technique based on regularization in regression is used to quantify the influence of several large-scale climate modes of variability on the basin-averaged manometric sea level. In most cases, we found consistency in the results using the three different datasets. The analysis indicates the NPGO (North Pacific Gyre Oscillation), AO (Arctic Oscillation), and AMO (Atlantic Multidecadal Oscillation) to be the most influential modes for the North Atlantic Ocean, Mediterranean Sea, and Arctic Ocean, respectively. This is the combined result of the barystatic sea level signature, cross-basin exchanges, and teleconnection patterns, as explained in detail in previous studies (Landerer and Volkov, 2013; Iglesias et al., 2018; Fang

et al., 2022). These results are useful as a reference for further fingerprinting technique applications and as a possible tool for statistical prediction of manometric variations.

The results provide a summary of the manometric sea level variability within the three basins investigated here and guide users in the choice of the specific product, depending on the region of interest. The overarching conclusions are that reanalyses when an ensemble mean of different systems is adopted, provide good performances in all basins; SLB performance is the most affected by observational sampling, and thus should be avoided in regions with poorly developed networks; gravimetry data provide realistic sub-seasonal and interannual variability, although long-term trends are less consistent than other datasets and the monthly uncertainty is the largest.

Further studies are needed to understand the different behavior of the datasets for certain aspects (e.g., the over-estimation of the Arctic Ocean manometric sea level trend by GRACE, or its under-estimation in the Mediterranean Sea), namely whether this is due to some intrinsic limitations of the data processing, or the different processes implied by the measurement techniques.

*Data availability*

See Table 1 for accessing the data through the associated DOI.

*Author contribution*

AS designed the analysis and wrote most of the text; GC and AS performed the analysis; JP coordinated the processing of the manometric sea level data from GRACE, revised the manuscript, and provided many comments on the use of the observation-based datasets and the fingerprinting technique; AB corrected in detail the Data and methods section; all authors contributed with data production and suggestions for improving the study.

*The authors declare no competing interests.*

*Acknowledgments*

The authors thank the OSR8 team (Karina von Schuckmann and Lorena Moreira Mendez) for their coordination efforts and suggestions to improve the quality of the original version of the manuscript. This work has been supported by the GLORAN-

Lot8 and the WAMBOR contracts of the Copernicus Marine Service.

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

| Product Ref. No | Product ID & Type | Data Access | Documentation |
|---|---|---|---|
| 1 | GLOBAL_REANALYSIS_PHY_001_031 (GREP), numerical models | EU Copernicus Marine Service Product (2022a) | QUID (Quality Information Document): Desportes et al. (2022)<br><br>PUM (Product User Manual): Gounou et al. (2022) |
| 2 | Barystatic and manometric from satellite gravimetry (LEGOS - MAGELLIUM) | Aviso Odatis webpage, 2023: doi: 10.24400/527896/a01-2023.011 | PUM (Product User Manual): https://www.aviso.altimetry.fr/fileadmin/documents/data/products/indic/WAMBOR-DT-009-MAG_CopernicusMarine_ServiceEvolution_PUM_v2.0.pdf |
| 3 | Barystatic and manometric from sea level budget (LEGOS - MAGELLIUM) | Aviso Odatis webpage, 2023: 10.24400/527896/a01-2023.012 | PUM (Product User Manual): https://www.aviso.altimetry.fr/fileadmin/documents/data/products/indic/WAMBOR-DT-009-MAG_CopernicusMarine_ServiceEvolution_PUM_v2.0.pdf |
| 4 | SEALEVEL_GLO_PHY_L4_MY_008_047, L4 reprocessed altimetry observations | EU Copernicus Marine Service Product (2022b) | QUID (Quality Information Document): Pujol et al. (2023)<br><br>PUM (Product User Manual): Pujol (2022) |

**Table 1. Product Table**

| Region | Trend | | | Seasonal amplitude | | | Interannual variability | | | Average uncertainty | | |
|---|---|---|---|---|---|---|---|---|---|---|---|---|
| | GRACE | SLB | GREP | GRACE | SLB | GREP | GRACE | SLB | GREP | GRACE | SLB | GREP |
| Arctic Ocean | 3.45 +/- 0.57 | 1.09 +/- 0.44 | 2.45 +/- 0.44 | 29.0 | 26.0 | 28.7 | 20.9 | 22.2 | 17.6 | 29.0 | 12.9 | 8.5 |
| North Atlantic Ocean | 2.67 +/- 0.23 | 3.24 +/- 0.16 | 1.81 +/- 0.18 | 14.2 | 10.7 | 14.4 | 6.0 | 6.6 | 6.1 | 29.9 | 20.8 | 8.0 |
| Mediterranean Sea | 0.87 +/- 0.65 | 2.44 +/- 0.50 | 1.93 +/- 0.46 | 31.5 | 25.5 | 30.0 | 27.8 | 29.2 | 20.0 | 31.8 | 11.8 | 13.1 |

Table 2. Manometric sea level diagnostics for the three basins considered in this study, calculated from the three datasets GREP (ensemble mean), GRACE, and SLB. The trend is calculated as a linear fit, with uncertainty found through bootstrapping. Seasonal amplitude and interannual variability are defined according to section 2.4. Average uncertainty is calculated from the gridpoint values. For GREP, it is given by the ensemble standard deviation. Units are mm yr$^{-1}$ for the trend, and mm for the other metrics.

| Region | Monthly timeseries | | Interannual timescale | | Seasonal timescale | |
|---|---|---|---|---|---|---|
| | GRACE | SLB | GRACE | SLB | GRACE | SLB |
| Arctic Ocean | 35% | 11% | 25% | 11% | 48% | 38% |
| North Atlantic Ocean | 56% | 80% | 71% | 79% | 34% | 85% |
| Mediterranean Sea | 4% | 19% | 1% | 11% | 8% | 37% |

Table 3. Percent of the regional manometric sea level variance explained by the global barystatic signal, also for the interannual and seasonal signals. The global barystatic signal is shown in Figure 1 as gray lines.

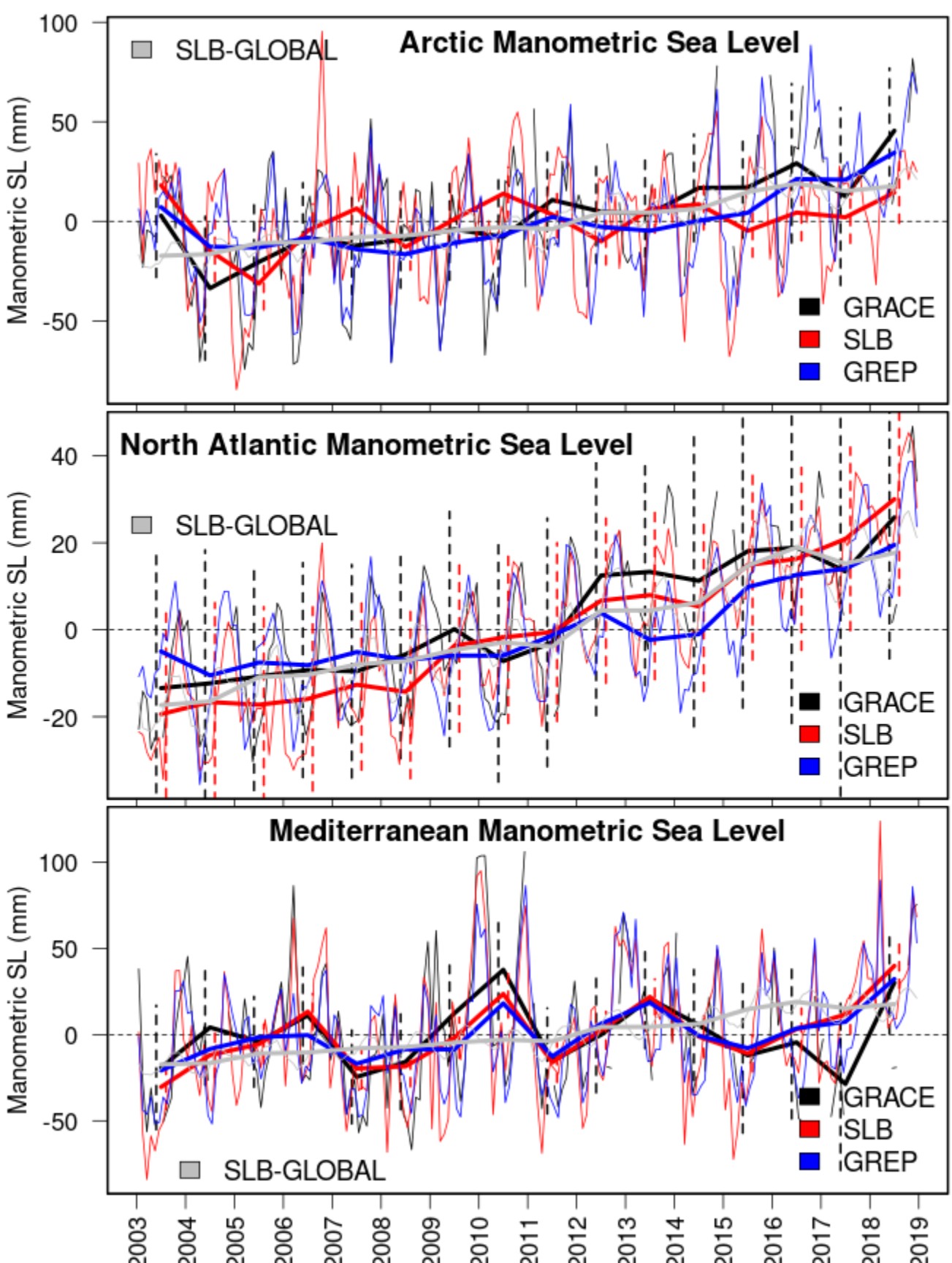

**Figure 1. Manometric sea level timeseries for the Arctic, Mediterranean, and North Atlantic basins. Both monthly (thin lines) and yearly (thick lines) means are shown for GRACE, SLB, and GREP. The global barystatic sea level (SLB method) is also added in gray. The North Atlantic Ocean is defined from 0°N to 67°N, and the Arctic Ocean from 67°N in the Atlantic Ocean to the Bering Strait. Dashed vertical lines correspond to the yearly uncertainty (for GRACE and SLB only; for GREP are not shown for sake of clarity, given their underestimated value due to the small ensemble size).**

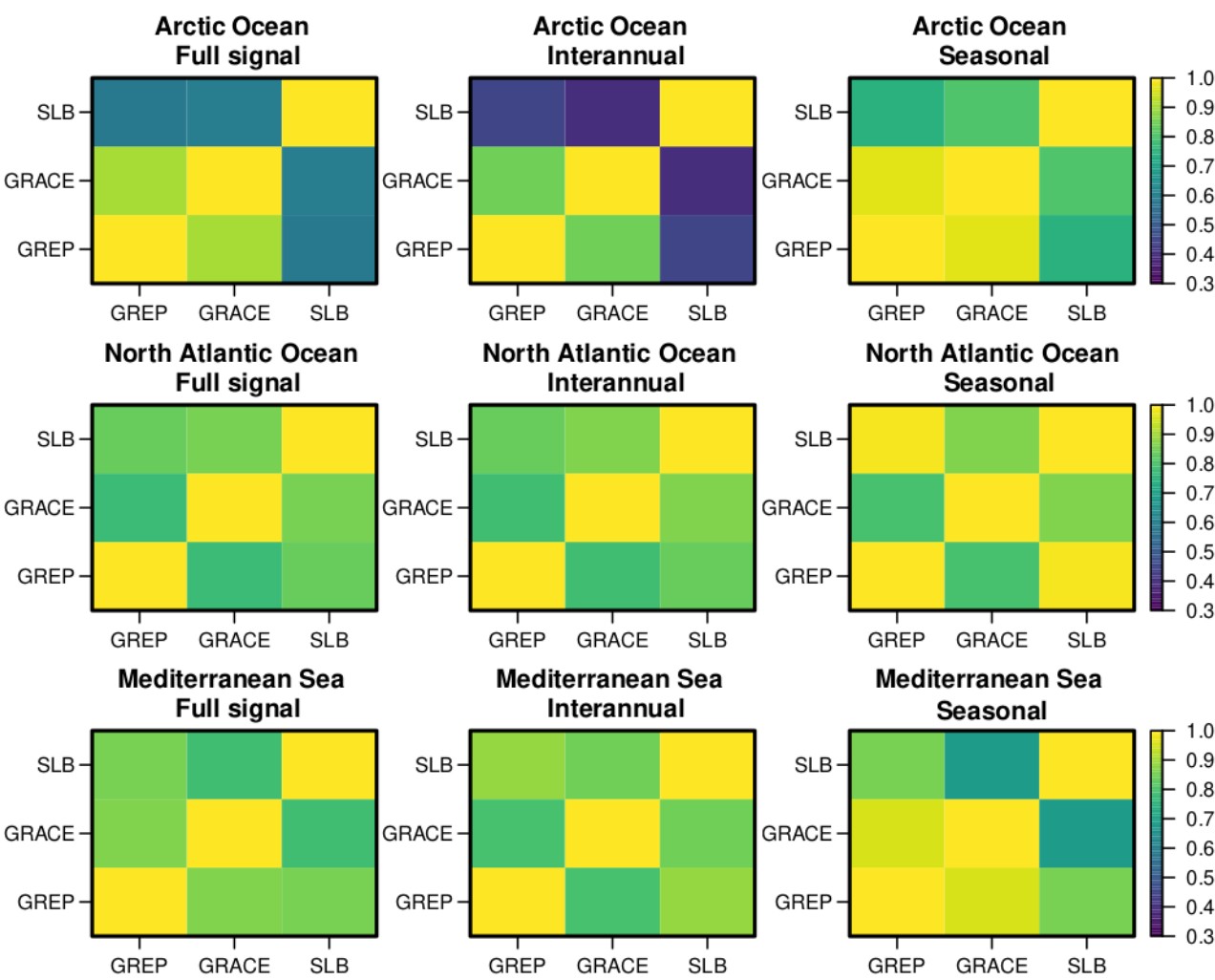

**Figure 2. Correlation matrix for the three datasets in the three ocean basins investigated in this study, for both the full, the interannual, and the seasonal signals. All values of correlation are statistically significant, at the 99% confidence level. Note that the correlation matrix is symmetric, but all terms are shown in any case for the sake of clarity; note also that the minimum correlation in the palette is 0.3, whereas the minimum correlation across all data shown is 0.39.**

605

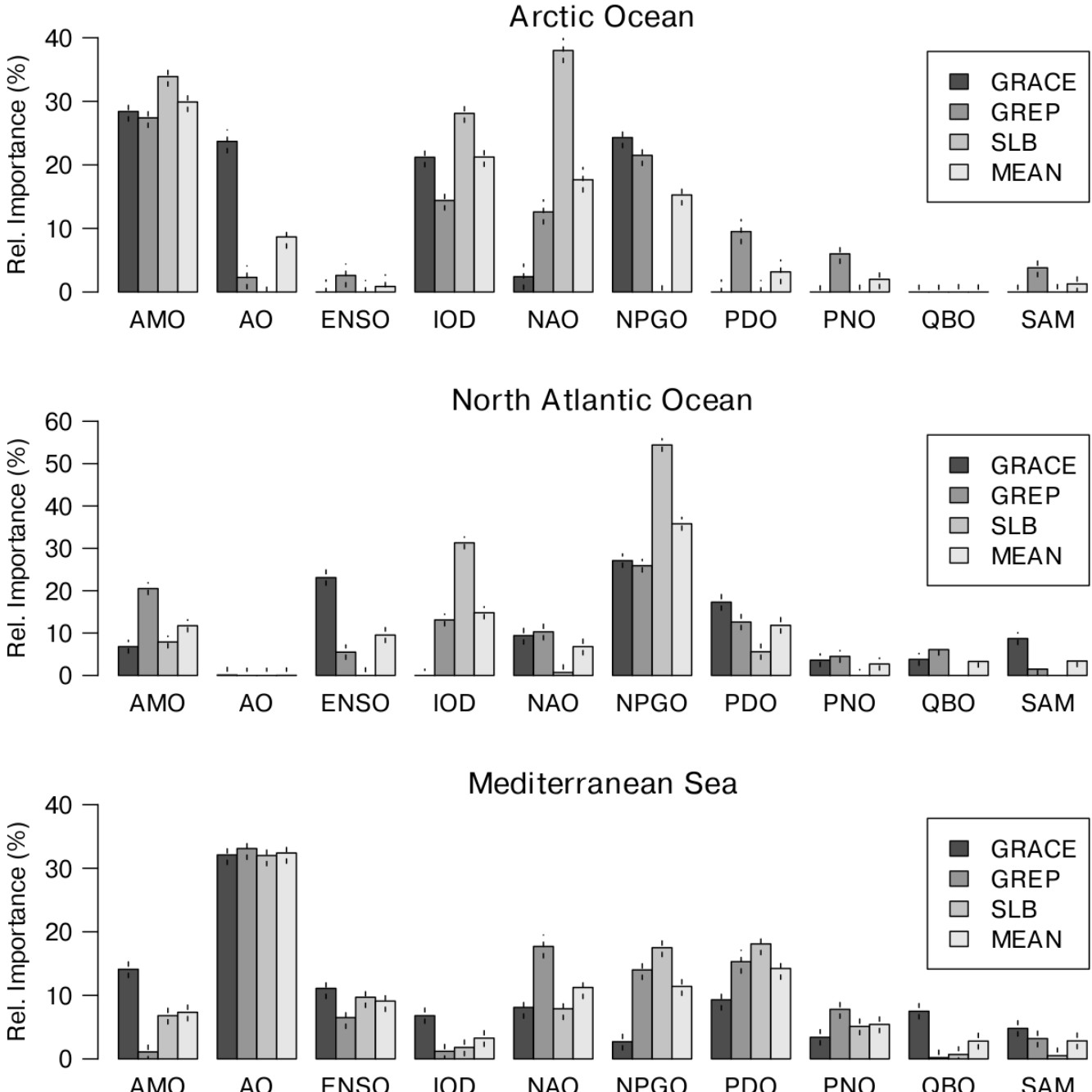

610

**Figure 3. Relative importance (defined in the text in section 2.4) of the selected climate indices for the manometric sea level in the three basins investigated in this study, using the three datasets GRACE, GREP, and SLB. Also shown is the mean of the relative importance over the three datasets (indicated as MEAN). The climate indices acronyms are as follows: AMO: Atlantic Multidecadal Oscillation; AO: Arctic Oscillation; ENSO: multivariate El Niño Southern**

615 **Oscillation; IOD: Indian Ocean Dipole; NAO: North Atlantic Oscillation; NPGO: North Pacific Gyre Oscillation; PDO: Pacific Decadal Oscillation; PNO: Pacific North American Oscillation; QBO: Quasi-Biennial Oscillation; SAM: Southern Annular Mode. Vertical bars indicate the regression coefficients' standard errors.**