# Peer review of "Variability of manometric sea level from reanalyses and observation-based products over the Arctic and North Atlantic Oceans and the Mediterranean Sea"

_State of the Planet, 2023_

## Author Comment (AC1)

*SP-2023-28 Submitted on 02 Aug 2023*
*"Variability of manometric sea level from reanalyses and observation-based products over the Arctic and North Atlantic Oceans and the Mediterranean Sea" by*
*Andrea Storto, Giulia Chierici, Julia Pfeffer, Anne Barnoud, Romain Bourdalle-Badie, Alejandro Blazquez, Davide Cavaliere, Benjamin Coupry, Marie Drevillon, Sebastien Fourest, Gilles Larnicol, and Chunxue Yang*
*Report: 8th edition of the Copernicus Ocean State Report (OSR8)*

**Reply to Anonymous Reviewer 1**

We thank the reviewer for the careful reading, the encouraging comments, and the many suggestions to improve the quality of the manuscript. We report below a point-by-point answer to the Reviewer's comments (Reviewer in bold font, our reply in regular font). Additionally, please note the limits in terms of the number of words and the number of figures for this type of submission, which we already exceeded in the original version; therefore we cannot add new figures/text but only improve/change/rearrange the original manuscript.
Finally, the GRACE dataset went through reprocessing, and we have replaced the previous dataset with the most recent one, which shows some non-negligible changes in the North Atlantic and Arctic regions.

**Main comments**

**This is a short and to the point paper, but perhaps it is a bit too short: I am missing some details and information that I think would make it stronger and more informative/appealing, as at the end, I'm left with the question: is there one method really better than the others? The authors write that 'The results are intended to (..) guide users in the choice of the specific product, depending on the region of interest' (L282-283), but to me it is not clear what the choice should then be based on, as with this information it is not possible to pick a 'best' approach, or is there something I've missed?**

We are adding a few sentences on this in the last section (say lines), also incorporating the comments from the second reviewer about reliability of the data in the Arctic region, observational sampling, etc.

**Uncertainties. There is very little attention to the spread in the results, and uncertainties are only sparingly mentioned or shown. For instance, Figure 1 (or any of the figures and most of the tables) shows no uncertainties, while this should be possible (?), given that for instance the GRACE dataset is an ensemble of 120 solutions. Including the uncertainties is essential to get a feeling for the consequences of using different methods in the manometric sea level in the different basins, and as it stands the three methods can only be compared very qualitatively.**

Adding uncertainties in Figure 1 would decrease the legibility of the figure, already very busy. Note also that Table 2 (and related discussion) already contains (last column) the time-averaged uncertainty for each dataset and basin. In the revised version of the manuscript, we aim to add the uncertainty bars on the yearly mean lines of Figure 1.

**Comparison to the global mean/total sea level change. Is it possible in figure 1 to also include (a panel showing) the global mean barystatic change for the three methods? (or at least GRACE and SLB, given the argumentation in l161?). Now showing only the global barystatic from SLB in Fig 1 feels a bit arbitrary as the reader does not know how similar (or different) these global time series are.In fact, showing the total sea level change (not only the manometric) for the global mean and the basins might be interesting too for reference, especially since for instance l.200 refers to the total change?**

As mentioned above, the length of the manuscript is limited, so we cannot add any more figures. The reviewer may refer to Barnoud et al., 2023b (reference available in the manuscript) for comparing the barystatic sea level changes from the SLB and GRACE methods. We will add a sentence on this in the revised version of the manuscript.

**Regional differences. Is it possible to include maps: how does the manometric signal vary spatially in these basins? I understand that time series are difficult, but the authors could for instance plot the linear manometric trend (mm/yr)?**

This is an interesting point; however, as we mentioned above, we already exceeded the length of the manuscript and the number of total figures. Unless the editor grants us the possibility of adding another figure, to remain within the manuscript limits we cannot add a figure. We will comment on the spatial distribution in the revised version in a new sentence, adding, however, "(not shown)".

**Figures. Please, can the figures be constructed in a colour-blind friendly way by choosing different colours (figs 1&2) and/or line styles (fig 1)? I'd suggest to change the colour bar of Fig 2 into a gradual one (choosing one colour which gets darker for higher correlation), as the colours now make it near impossible to interpret this figure. (see https://www.nature.com/articles/s41467-020-19160-7 for reasons why the rainbow scale is not a good scale to use). Alternatively: wouldn't it make sense to provide this fig3 information in a table format, so that uncertainties can also be included? Fig 1; Would it make sense to plot the linear trends in figure 1? (it may become too busy though). Fig 2; given that these correlations are mirrored, wouldn't it make sense to only show the half matrices, as basically one only needs the three blocks in the upper left corner of each correlation plot.  Fig 3; can uncertainties whiskers be included on the bars?**

Thanks for the suggestion, we will replot the figures in a color-blind-friendly palette. Regarding the additions: figure 1 is already too busy, especially if we add the uncertainty bars. Figure 2: we can explicitly state that the correlation matrices are symmetric by construction; however, we already tried to plot half matrix only, and the plot is less aesthetically appealing than plotting the full matrix. Figure 3: yes we will add the uncertainty.

**Minor comments**

**Is there a specific reason for focusing on these three basins? The data covers the global ocean, doesn't it?**

We chose these basins as a compromise between geographical interests (basins of interest for the European communities and, thus, the Copernicus Marine Service, excluding however too small basins - Black and Baltic Seas, etc. - which won't be enough constrained by the observing networks used, and for which the recourse to regional modeling systems would be more appropriate). We will add a short sentence on this.

**L 85. 'assessing the multi-method mean signal' – I don't think this is done in the paper? I could only find this for the separate methods?**

Thanks for spotting this inconsistency. Indeed, this objective was planned in a preliminary version but has not been treated in the present manuscript, and will therefore be removed in the revised version.

**L184 'significantly different' – is this statistical significance?**

The basins show many statistically significantly different metrics, but here it was meant in a more general (not statistical) sense, so we remove "significant" for clarity.

**L186-188; 'except during the first and last years'? ; is it only due to the final year that the trend is this high? How sensitive is the trend to those first and last years?**

Thanks for pointing this out. Indeed, the bootstrapping technique used to quantify the trend uncertainty removes part of the timeseries, and thus exactly quantifies the sensitivity of the trend to individual years. We add a sentence on this to explicitly point it out.

**L190 add a cross-ref to Table 3 here**

It will be added in the revised version

**L200 'the global barystatic signal'?**

The barystatic is defined as the mass component of the global mean sea level changes (Gregory et al., 2019). Therefore, referring to the global barystatic is a repetition (tautology). We thus prefer to use one (global signal) or the other (barystatic signal).

**l203 – unclear what 'the total trend' is: is this the total barystatic trend, and is it in the basin or the global mean? How can the trend in a basin 'explain' a total trend? (the other way around sounds more logical?)**

"total sea level trend" means the SSH trend (manometric plus steric) as seen by altimetry. We will clarify this point in the revised version.

**L212 – 'generally': in the NA and Medi, the correlations between GRACE and other datasets are always lower than for the SLB-GREP combo, isn't it?**

Thanks, you are right. We will modify the sentence accordingly, removing the adverb "generally".

---

## Author Comment (AC2)

*SP-2023-28 Submitted on 02 Aug 2023*
*"Variability of manometric sea level from reanalyses and observation-based products over the Arctic and North Atlantic Oceans and the Mediterranean Sea" by*
*Andrea Storto, Giulia Chierici, Julia Pfeffer, Anne Barnoud, Romain Bourdalle-Badie, Alejandro Blazquez, Davide Cavaliere, Benjamin Coupry, Marie Drevillon, Sebastien Fourest, Gilles Larnicol, and Chunxue Yang*
*Report: 8th edition of the Copernicus Ocean State Report (OSR8)*

**Reply to Reviewer 2**

We thank the reviewer for the careful reading, the encouraging comments, and the many suggestions to improve the quality of the manuscript. We report below a point-by-point answer to the Reviewer's comments (Reviewer in bold font, our reply in regular font). Additionally, please note the limits in terms of the number of words and the number of figures for this type of submission, which we already exceeded in the original version; therefore we cannot add new figures/text but only improve/change/rearrange the original manuscript.
Finally, the GRACE dataset went through reprocessing, and we have replaced the previous dataset with the most recent one, which shows some non-negligible changes in the North Atlantic and Arctic regions.

**Comments**

**I have serious concerns on the calculations over the Arctic Ocean from the SLB method (primarily from the Argo data), with slightly lesser concerns about the North Atlantic study.**
**How do the authors calculate steric anomalies in the Arctic when there are only small numbers of floats in this region???? As someone who is currently working on analyzing individual Argo float data for a project and not just using analyzed grids, I can assure the authors that there are not sufficient observations in the Arctic to even begin to do the calculations they are attempting. Most of the analysis grids cut-off the data at 65° because of this. If there are "values" in the grid cells, they must be from a climatology or VERY limited data and extrapolation. Unless the authors can justify that there are sufficient Argo observations in the Arctic to support their calculations, I cannot accept that any calculation of ocean mass based on altimetry - Argo data is credible in the Arctic Ocean.**

We thank the reviewer for this comment; indeed, the Arctic region is affected by large uncertainties for the SLB product, due to the poor observational sampling. Our approach for the revised version is, however, to i) explicitly mention these limitations for the SLB product (the other two being less affected, as reanalysis bears information on the meridional transports and atmospheric forcing, and GRACE is not necessarily affected besides issues with satellite footprint geometry and leakage); ii) discuss in more details the uncertainty of the products and compare it between the products, as also suggested by Reviewer 1. Please note, that the focus here is not to show only the SLB product but to discuss the advantages and disadvantages of the three datasets on the study basins. That is why we do not want to exclude SLB in the Arctic, but better mention and quantify its limitations at high latitudes.

**There may be some validity in the using the ocean state estimate, but this is also limited by the altimetry data problems over the Arctic due to inclinations of the satellites, sea ice, etc. In the manuscript between lines 185-195, the authors point out that the trend in ocean mass using the GREP method (the ocean reanalysis steric correction) is 6.2 mm/year, compared to 2.5 mm/year for the gravimetry (or only a little over the global mean rate). They comment later: "Note that the GRACE-derived trend is likely too large, as it exceeds the altimetry-based total sea level trend of 2.9 mm yr-1, although the latter is characterized by significant under-sampling at high latitudes and ice-covered regions". I have to assume this later statement is referring to the GREP estimate, not GRACE. And if it is 6.2 mm/year, that means an enormous negative steric change over the Arctic (3 mm - steric = 6.2, means steric = -3.2 mm/year. This would have to be caused by either a cooling or salt gain, which doesn't really make physical sense based on observations of the Arctic warming and freshening. This does not encourage one to trust either the GREP or the ARGO-based SLB estimate, and I suspect for the same reasons -- there simply are not enough T/S measurements there to support the calculation. IMO, the entire Arctic analysis should be stricken because the data sets being used are not adequate to measure what the authors want. If they want to include it, they need to do a much better job of describing the limitations of the various data and models in the Arctic.**

Please see above for the issue of the Arctic Ocean data reliability. We will make this point much more clear in the revised version, but we still believe there is value in assessing and comparing the manometric sea level in the Arctic, eventually pointing to the weaknesses (these datasets are however state-of-the-science datasets that are broadly used also for Arctic studies, regardless of the observational undersampling).

Additionally, it is important to note here that we do not expect the budget to close at the regional scale and that large errors may affect any of the components of the budget at the regional scale (altimetry, grace, and Argo), especially over the Arctic where there are known issues with sampling both for the in-situ and altimetry datasets. The new version of the GRACE dataset over the Arctic shows a trend that will likely be less than 2.5 mm/yr (we still need to redo the computation with the same definition as in the manuscript).

The reviewer is of course correct, that we confused GRACE with GREP in the discussion

We will add all these points in the revised version of the manuscript.

**The analysis in the North Atlantic and Med. Sea is better, because the data can support this, but I am troubled by the fact the authors did not use the salinity data in the Argo-based method. They do this because of a small residual drop post 2016 that appears in global halsosteric sea level variations. While this is a legitimate concern for global studies, it is not for regional studies. There can be large, real halosteric (salinity) fluctuations that are balanced by a compensating temperature change. This is known as "density compensation" and is a common feature in T/S data where water masses are being mixed, there are large fronts, eddies, and deep convection -- all common features of the North Atlantic. I can assure you that if you look at the T/S grids (or profiles) you will see these features all over the Atlantic and Med. Sea and that they can be quite large -- tens of cm of halosteric/thermosteric sea level change, that when added cancel so the steric change is small. Such an event happened in the late 70s called the "Great salinity Anomaly" and based on what I have been observing in the Argo profiles, something similar has been happening over the last several years. I haven't seen any**

**papers on it yet, but my point is that the authors should NOT exclude salinity from their calculations on regional estimates. Yes, there may a small global drift, but by ignoring salinity they are eliminating signals that are tens of cm that will cancel some of the thermometric variations. These will not necessarily average to zero.I suggest the authors recompute their Argo-based SLB estimate including salinity, and just note that the trend may be a little off at the end of the record because of the apparent salinity drift.**

Please note, that there has been a misunderstanding here, as probably the text in the original manuscript was not clear enough. Indeed, we only neglect the halosteric contribution for the calculation of the global mean (i.e. the barystatic). In all regional time series - on which the manuscript is based - we do consider the full steric signal, including both the thermosteric and halosteric components, for the SLB product. Therefore, we thank the reviewer for his extensive comments, and we are sorry for the misunderstanding, but these do not apply to the data used in the manuscript. In the revised version of the manuscript, we have now stressed that the manometric estimate based on the SLB method is calculated as the difference between the total sea level changes from altimetry and the full steric changes from Argo, including the halosteric variations.

**My final major concern is on the analysis of the relationship with various climate indices. This section is so short, that I cannot fully understand how the relationships were established or if the climate indices were smoothed in any way. For instance, the NAO has a lot of short-term (month-month) variability, while the AMO has a large 60-year oscillation which will correlate with the trend over a twenty year period. That's not really good evidence of a relationship. But not knowing exactly what was done, and how the percentages were computed, I cannot judge this. Unless the authors choose to expand this with a more thorough description of the analysis, I cannot support it being included.**

In the revised version we will add some details on the calculation and add some discussions on the climate modes, although the length of the paper is very limiting. Data used for these calculations are monthly raw means without any low-pass filtering, similar to many other works focussing on the climate mode fingerprints on sea level. We have followed this strategy without arbitrarily filtering the data, in the multivariate regression framework.

**There are some more minor comments, all of which should be easy to fix:**

**1. On the discussion of the Boussinesq approximation: authors should add "cannot represent the steric expansion…" They can (and do) measure the non-global parts quite well.**

Thanks for this point, we include it in the revised version to stress that only the global steric signal.

**2. It is true that reanalysis models "make barystatic and manometric terms often unrealistic" is true. But in most cases they are also non-existent! This should be added.**

This point is not clear. All reanalysis models are forced by atmospheric reanalyses, and the freshwater cycle implied by this forcing is not balanced, meaning that the barystatic and manometric terms are not realistic by construction (but existent). Maybe we missed the reviewer's point; therefore, we prefer to leave the manuscript unchanged.

**3. There are some state estimates that have begun to assimilate gravimetry and do have reasonable barystatic variations: ECCO_v4r4 is one. Rui Ponte recently used it for diagnosing the freshening of the ocean in a paper in GRL.**

We will add the correct reference for ECCO to show reasonable barystatic variations.

**4. "A linear trend of 0.12 ± 0.03 mm yr-1 is added to consider the contribution of the deep ocean to thermosteric sea level changes (Chang et al., 2019)." This is a value for the global average, and appropriate only for GLOBAL steric estimates. This is driven primarily by deep warming in the Southern Ocean, with lesser signals in the North Atlantic and NO evidence (or data) for such a trend in the Arctic. Please remove this "correction" for these data and merely comment on the potential of deep warming signals that are not accounted for and give some ranges -- and not just a global value! In fact, the authors should be able to analyze this somewhat with their ocean reanalysis output. What does it say?**

We will remove the deep ocean correction in the revised version of the manuscript and acknowledge the fact that steric signals from the deep ocean are not accounted for. However, we are not able to give an order of magnitude for the potential effects of the deep ocean warming at the regional scale. There is currently not enough data to do so. Ocean reanalyses are poorly constrained as well in the deep ocean. Any range of values given for the steric contribution of the deep ocean at the basin scale would be affected by large uncertainties.

**5. "Explained variance, as percent R2 coefficient, is used to quantify how much of the regional signal is explained by the global barystatic signal due to fast barotropic motion." R2 (based on squaring the correlation) is NOT explained variance unless the data being compared have exactly the same variance. In this case, they likely do not. Better to compare variances: PVE = 1 – var(resid)/var(orig), where resid is the original time series minus the global barystatic signal. This is really variance explained.**

Thanks, we will correct the reference to the explained variance versus R2 coefficient as suggested by the Reviewer (in both the methods and results sections).

---

## Author Response (AR1)

*SP-2023-28 Submitted on 02 Aug 2023*
*Reply to the reviewers.*
*"Variability of manometric sea level from reanalyses and observation-based products over the Arctic and North Atlantic Oceans and the Mediterranean Sea" by Andrea Storto, et al.*

We thank the reviewers for the careful reading, the encouraging comments, and the many suggestions to improve the quality of the manuscript. We report below a point-by-point answer to the Reviewer's comments (Reviewer in bold font, our reply in regular font). Additionally, please note the limits in terms of the number of words and the number of figures for this type of submission, which we already exceeded in the original version; therefore, we cannot add new figures/text but only improve/change/rearrange the original manuscript.

Finally, the GRACE dataset went through reprocessing, and we have replaced the previous dataset with the most recent one, which shows some non-negligible changes in the North Atlantic and Arctic regions. Additionally, the SLB dataset is also adjusted to account for the deep ocean correction that is not used any longer at the regional scale. With such reprocessing, the consistency of the three datasets has even improved, and a few critical issues (e.g. large Arctic trend) have been mitigated.

**Reply to Anonymous Reviewer 1**

**Main comments**

**This is a short and to the point paper, but perhaps it is a bit too short: I am missing some details and information that I think would make it stronger and more informative/appealing, as at the end, I'm left with the question: is there one method really better than the others? The authors write that 'The results are intended to (..) guide users in the choice of the specific product, depending on the region of interest' (L282-283), but to me it is not clear what the choice should then be based on, as with this information it is not possible to pick a 'best' approach, or is there something I've missed?**

We added a sentence on this in the last section (lines 300-305), also incorporating the comments from the second reviewer about the reliability of the data in the Arctic region, observational sampling, etc.

**Uncertainties. There is very little attention to the spread in the results, and uncertainties are only sparingly mentioned or shown. For instance, Figure 1 (or any of the figures and most of the tables) shows no uncertainties, while this should be possible (?), given that for instance the GRACE dataset is an ensemble of 120 solutions. Including the uncertainties is essential to get a feeling for the consequences of using different methods in the manometric sea level in the different basins, and as it stands the three methods can only be compared very qualitatively.**

Adding uncertainties in Figure 1 would decrease the legibility of the figure, already very busy. Note also that Table 2 (and related discussion) already contains (last column) the time-averaged uncertainty for each dataset and basin. In the revised version of the manuscript, we added the uncertainty bars on the yearly mean lines of Figure 1.

**Comparison to the global mean/total sea level change. Is it possible in figure 1 to also include (a panel showing) the global mean barystatic change for the three methods? (or at least GRACE and SLB, given the argumentation in l161?). Now showing only the global barystatic from SLB in Fig 1 feels a bit arbitrary as the reader does not know how similar (or different) these global time series are. In fact, showing the total sea level change (not only the manometric) for the global mean and the basins might be interesting too for reference, especially since for instance l.200 refers to the total change?**

As mentioned above, the length of the manuscript is limited, so we cannot add any more figures. The reviewer may refer to Barnoud et al., 2023b (the reference is available in the manuscript) for comparing the barystatic sea level changes from the SLB and GRACE methods. We added a sentence on this in the revised version of the manuscript (section 2.4)

**Regional differences. Is it possible to include maps: how does the manometric signal vary spatially in these basins? I understand that time series are difficult, but the authors could for instance plot the linear manometric trend (mm/yr)?**

This is an interesting point; however, as we mentioned above, we already exceeded the length of the manuscript and the number of total figures. Unless the editor grants us the possibility of adding another figure, to remain within the manuscript limits we cannot add a figure. We will comment on the spatial distribution in the revised version in a new sentence, adding, however, "(not shown)".

**Figures. Please, can the figures be constructed in a colour-blind friendly way by choosing different colours (figs 1&2) and/or line styles (fig 1)? I'd suggest to change the colour bar of Fig 2 into a gradual one (choosing one colour which gets darker for higher correlation), as the colours now make it near impossible to interpret this figure. (see https://www.nature.com/articles/s41467-020-19160-7 for reasons why the rainbow scale is not a good scale to use). Alternatively: wouldn't it make sense to provide this fig3 information in a table format, so that uncertainties can also be included? Fig 1; Would it make sense to plot the linear trends in figure 1? (it may become too busy though). Fig 2; given that these correlations are mirrored, wouldn't it make sense to only show the half matrices, as basically one only needs the three blocks in the upper left corner of each correlation plot. Fig 3; can uncertainties whiskers be included on the bars?**

Thanks for the suggestion, we have replotted the figures in a color-blind-friendly palette. Regardingthe additions: figure 1 is already too busy, especially if we add the uncertainty bars. Figure 2:we now explicitly state that the correlation matrices are symmetric by construction; however, we already tried to plot half matrix only, and the plot is less aesthetically appealing than plottingthe full matrix. Figure 3: we have added the uncertainty.

**Minor comments**

**Is there a specific reason for focusing on these three basins? The data covers the global ocean, doesn't it?**

We chose these basins as a compromise between geographical interests (basins of interest for the European communities and, thus, the Copernicus Marine Service, excluding however too small basins - Black and Baltic Seas, etc. - which won't be enough constrained by the observing networks used, and for which the recourse to regional modeling systems would be more appropriate). We added a short sentence on this in the first paragraph of the Summary & Conclusions.

**L 85. 'assessing the multi-method mean signal' – I don't think this is done in the paper?I could only find this for the separate methods?**

Thanks for spotting this inconsistency. Indeed, this objective was planned in a preliminary version but has not been treated in the present manuscript and is therefore removed in the revised version.

**L184 'significantly different' – is this statistical significance?**

The basins show many statistically significantly different metrics, but here it was meant in a more general (not statistical) sense, so we remove "significant" for clarity.

**L186-188; 'except during the first and last years'? ; is it only due to the final year that the trend is this high? How sensitive is the trend to those first and last years?**

Thanks for pointing this out. Indeed, the bootstrapping technique used to quantify the trend uncertainty removes part of the timeseries, and thus exactly quantifies the sensitivity of the trend to individual years. We added a sentence on this to explicitly point it out, in section 2.4.

**L190 add a cross-ref to Table 3 here**

Added in the revised version.

**L200 'the global barystatic signal'?**

Corrected.

**l203 – unclear what 'the total trend' is: is this the total barystatic trend, and is it in the basin or the global mean? How can the trend in a basin 'explain' a total trend? (the other way around sounds more logical?)**

"total sea level trend" means the SSH trend (manometric plus steric) as seen by altimetry, and not the global as the reviewer imagined. We clarified this point in the revised version.

**L212 – 'generally': in the NA and Medi, the correlations between GRACE and other datasets are always lower than for the SLB-GREP combo, isn't it?**

Thanks, you are right. We modified the sentence accordingly, removing the adverb "generally".

**Reply to Reviewer 2**

**Comments**

**I have serious concerns on the calculations over the Arctic Ocean from the SLB method (primarily from the Argo data), with slightly lesser concerns about the North Atlantic study.**
**How do the authors calculate steric anomalies in the Arctic when there are only small numbers of floats in this region???? As someone who is currently working on analyzing individual Argo float data for a project and not just using analyzed grids, I can assure the authors that there are not sufficient observations in the Arctic to even begin to do the calculations they are attempting. Most of the analysis grids cut-off the data at 65° because of this. If there are "values" in the grid cells, they must be from a climatology or VERY limited data and extrapolation. Unless the authors can justify that there are sufficient Argo observations in the Arctic to support their calculations, I cannot accept that any calculation of ocean mass based on altimetry - Argo data is credible in the Arctic Ocean.**

We thank the reviewer for this comment; indeed, the Arctic region is affected by large uncertainties for the SLB product, due to the poor observational sampling. Our approach for the revised version was to i) explicitly mention these limitations for the SLB product (the other two being less affected, as reanalysis bears information on the meridional transports and atmospheric forcing, and GRACE is not necessarily affected besides issues with satellite footprint geometry and leakage); ii) discuss in more details the uncertainty of the products and compare it between the products, as also suggested by Reviewer 1. Please note, that the focus here is not to show only the SLB product but to discuss the advantages and disadvantages of the three state-of-the-art datasets, for the selected basins. That is why we do not want to exclude SLB in the Arctic, but better mention and quantify its limitations at high latitudes.

**There may be some validity in the using the ocean state estimate, but this is also limited by the altimetry data problems over the Arctic due to inclinations of the satellites, sea ice, etc. In the manuscript between lines 185-195, the authors point out that the trend in ocean mass using the GREP method (the ocean reanalysis steric correction) is 6.2 mm/year, compared to 2.5 mm/year for the gravimetry (or only a little over the global mean rate). They comment later: "Note that the GRACE-derived trend is likely too large, as it exceeds the altimetry-based total sea level trend of 2.9 mm yr-1, although the latter is characterized by significant under-sampling at high latitudes and ice-covered regions". I have to assume this later statement is referring to the GREP estimate, not GRACE. And if it is 6.2 mm/year, that means an enormous negative steric change over the Arctic (3 mm - steric = 6.2, means steric = -3.2 mm/year. This would have to be caused by either a cooling or salt gain, which doesn't really make physical sense based on observations of the Arctic warming and freshening. This does not encourage one to trust either the GREP or the ARGO-based SLB estimate, and I suspect for the same reasons -- there simply are not enough T/S measurements there to support the calculation. IMO, the entire Arctic analysis should be stricken because the data sets being used are not adequate to measure what the authors want. If they want to include it, they need to do a much better job of describing the limitations of the various data and models in the Arctic.**

Please see above for the issue of the Arctic Ocean data reliability. We clarified this point in the revised version, but we still believe there is value in assessing and comparing the manometric

sea level in the Arctic, eventually pointing to the weaknesses (these datasets are however state-of-the-science datasets that are broadly used also for Arctic studies, regardless of the observational under-sampling).

Additionally, it is important to note here that we do not expect the budget to close at the regional scale and that large errors may affect any of the components of the budget at the regional scale (altimetry, grace, and Argo), especially over the Arctic where there are known issues with sampling both for the in-situ and altimetry datasets. The new version of the GRACE dataset over the Arctic shows a trend still quite large but not as unrealistic as before; masks are not exactly the same so the comparison with altimetry is only qualitative.

The large trend is from GRACE and not GREP, and we added explicitly that the dataset is not built to close the regional or basin-scale budget.

**The analysis in the North Atlantic and Med. Sea is better, because the data can support this, but I am troubled by the fact the authors did not use the salinity data in the Argo-based method. They do this because of a small residual drop post 2016 that appears in global halsosteric sea level variations. While this is a legitimate concern for global studies, it is not for regional studies. There can be large, real halosteric (salinity) fluctuations that are balanced by a compensating temperature change. This is known as "density compensation" and is a common feature in T/S data where water masses are being mixed, there are large fronts, eddies, and deep convection -- all common features of the North Atlantic. I can assure you that if you look at the T/S grids (or profiles) you will see these features all over the Atlantic and Med. Sea and that they can be quite large -- tens of cm of halosteric/thermosteric sea level change, that when added cancel so the steric change is small. Such an event happened in the late 70s called the "Great salinity Anomaly" and based on what I have been observing in the Argo profiles, something similar has been happening over the last several years. I haven't seen anypapers on it yet, but my point is that the authors should NOT exclude salinity from theircalculations on regional estimates. Yes, there may a small global drift, but by ignoringsalinity they are eliminating signals that are tens of cm that will cancel some of the thermometric variations. These will not necessarily average to zero.I suggest the authors recompute their Argo-based SLB estimate including salinity, and just note that the trend may be a little off at the end of the record because of the apparent salinity drift.**

Please note, that there has been a misunderstanding here, as probably the text in the original manuscript was not clear enough. Indeed, we only neglect the halosteric contribution for the calculation of the global mean (i.e. the barystatic). In all regional time series - on which the manuscript is based - we do consider the full steric signal, including both the thermosteric and halosteric components, for the SLB product. Therefore, we thank the reviewer for his extensive comments, and we are sorry for the misunderstanding, but these do not apply to the data used in the manuscript. In the revised version of the manuscript, we have now stressed that the manometric estimate based on the SLB method is calculated as the difference between the total sea level changes from altimetry and the full steric changes from Argo, including the halosteric variations. Now clarified in section 2.2.

**My final major concern is on the analysis of the relationship with various climate indices. This section is so short, that I cannot fully understand how the relationships were established or if the climate indices were smoothed in any way. For instance, the NAO has a lot of short-term (month-month) variability, while the AMO has a large 60-year oscillation which will correlate with the trend over a twenty-year period. That's not really good evidence of a relationship. But not knowing exactly what was done, and**

**how the percentages were computed, I cannot judge this. Unless the authors choose to expand this with a more thorough description of the analysis, I cannot support it being included.**

In the revised version we add some details on the calculation and some discussions on the climate modes, although the length of the paper is very limiting, and we cannot add many details. Data used for these calculations are monthly raw means without any low-pass filtering, like many other works focusing on the climate mode fingerprints on sea level (Pfeffer et al., 2022, and references therein). We have followed this strategy without arbitrarily filtering the data, in the multivariate regression framework. Details were added in section 2.4 and 3.

**There are some more minor comments, all of which should be easy to fix:**

**1. On the discussion of the Boussinesq approximation: authors should add "cannot represent the steric expansion…" They can (and do) measure the non-global parts quite well.**

Thanks for this point, we include it in the revised version to stress that only the global steric signal.

**2. It is true that reanalysis models "make barystatic and manometric terms often unrealistic" is true. But in most cases they are also non-existent! This should be added.**

This point is not clear. All reanalysis models are forced by atmospheric reanalyses, and the freshwater cycle implied by this forcing is not balanced, meaning that the barystatic and manometric terms are not realistic by construction (but existent). Maybe we missed the reviewer's point; therefore, we prefer to leave the manuscript unchanged in these regards.

**3. There are some state estimates that have begun to assimilate gravimetry and do have reasonable barystatic variations: ECCO_v4r4 is one. Rui Ponte recently used it for diagnosing the freshening of the ocean in a paper in GRL.**

We added the correct reference for ECCO, as an example (the only one) of data assimilation systems exploiting gravimetry data and showing reasonable barystatic variations.

**4. "A linear trend of 0.12 ± 0.03 mm yr-1 is added to consider the contribution of the deep ocean to thermosteric sea level changes (Chang et al., 2019)." This is a value for the global average, and appropriate only for GLOBAL steric estimates. This is driven primarily by deep warming in the Southern Ocean, with lesser signals in the North Atlantic and NO evidence (or data) for such a trend in the Arctic. Please remove this "correction" for these data and merely comment on the potential of deep warming signals that are not accounted for and give some ranges -- and not just a global value! In fact, the authors should be able to analyze this somewhat with their ocean reanalysis output. What does it say?**

Thanks for pointing this out. We have removed the deep ocean correction in the revised version of the data and manuscript, and we now acknowledge the fact that steric signals from the deep ocean are not accounted for. However,we are not able to give an order of magnitude for the potential effects of the deep ocean warming at the regional scale. The Magellium team has tried to use the ECMWF reanalysis ORAS5 to estimate the deep ocean warming, but this turned out to be unrealistic in several regions, and therefore the deep ocean warming has not

been accounted for anywhere. There is currently not enough data to do so. Ocean reanalyses are poorly constrained as well in the deep ocean. Any range of values given for the steric contribution of the deep ocean at the basin scale would be affected by large uncertainties, so we just neglect its effect.

**5. "Explained variance, as percent R2 coefficient, is used to quantify how much of the regional signal is explained by the global barystatic signal due to fast barotropic motion." R2 (based on squaring the correlation) is NOT explained variance unless the data being compared have exactly the same variance. In this case, they likely do not. Better to compare variances: PVE = 1 – var(resid)/var(orig), where resid is the original time series minus the global barystatic signal. This is really variance explained.**

Thanks, we have corrected the way the explained variance is defined and calculated, following the reviewer's suggestions. The results are slightly different, although also SLB and GRACE timeseries have been reprocessed, although most conclusions hold.

---

## Author Response (AR2)

*SP-2023-28 Submitted on 02 Aug 2023*
*Reply to the reviewers. Second round of reviews*
*"Variability of manometric sea level from reanalyses and observation-based products over the Arctic and North Atlantic Oceans and the Mediterranean Sea" by Andrea Storto, et al.*

We thank the reviewers for the careful reading, the encouraging comments, and the many suggestions to improve the quality of the manuscript. We appreciate the very careful checks of the reviewers that helped us spot some inconsistencies in the text, introduced during the revision and not identified earlier. We report below a point-by-point answer to the Reviewer's comments (Reviewer in bold font, our reply in regular font).

**Reply to Anonymous Reviewer 2**

**Main comments**

**The manuscript uses several observational and reanalysis datasets to describe and explain manometric sea-level variations in the Arctic Ocean, the North Atlantic Ocean, and the Mediterranean Sea. In addition, the manuscript reports strengths and limitations of the datasets used for the analysis. I would like to thank the authors for submitting this contribution. I found the manuscript interesting. However, I recommend a major revision of the manuscript before it is accepted for publication.**
Thanks for the comments

**Minor Issues**
**--------------**
**I would recommend the authors to check the text for:**
**• missing articles (e.g., "are applied to GRACE solutions" in L107)**
**• missing hyphens (e.g., "gravimetry based" in L63, or "in situ measurements" in L114/115)**
**• typos (e.g., "wet troposphere correction" instead of "wet tropospheric correction" in L122)**
**• missing words (e.g., "which covers from 1993 to 2019" in L145)**
**• repeated words (e.g., "changes" in L196)**
**These are just small mistakes, but they should be removed before the manuscript is accepted.**
Thanks. We have corrected all the points above, plus a few more small changes while re-reading the manuscript to improve the language.

**Abstract**
**---------**
**The acronym GREP is used without it being previously defined in the abstract.**
Thanks, GREP is now introduced with its acronymic meaning

**Short summary**
**-----------------**
**It is maybe better to find an alternative to "three different techniques". I am not sure we can refer to "reanalyses, gravimetry, and altimetry in combination with in-situ observations" as techniques.**
We changed techniques to methods.

**Data and methods**
* * *
**Section 2.4**
* * *
**1) In L159/160, the authors state: "… where the interannual signal is the timeseries to which the monthly climatology has been subtracted, and the subannual the residual part." I am not sure I understood this approach correctly. It seems to me that this method does not allow the authors to extract the subannual and the interannual components of a timeseries. In fact, the residual is the monthly climatology and, as such, it should mostly correspond to the seasonal cycle. If this were the case, the authors should avoid referring to the residual variability as 'subannual' because this term can be misleading.**

We thank the reviewer for pointing out this sentence; indeed, we used subannual basically as a synonym of seasonal, but we recognize it could be misleading and substituted all occurrences of "subannual" with "seasonal".

**2) There seems to be an inconsistency between what written in Section 2.4 and what written in the caption of Table 2 in relation to how the seasonal amplitude is computed. In L169, the authors write that "Seasonal amplitude is defined by fitting the monthly data to a sinusoidal curve". However, the caption of Table 2 states that "Seasonal amplitude stems from fitting the detrended timeseries to a sinusoidal line". Did the authors fit the sinusoidal curve to the detrended timeseries or to the original ones?**

Thanks for pointing this out, indeed was a typo between the two versions of the manuscript, as we re-did the calculation. We fit into a line with a trend term and a sinusoidal term and modified the text for clarity. For better clarity, we now stress that the fitting curve has a trend a seasonal components. To avoid redundancy, we refer to section 2.4 in Table 2's caption.

**3) In L169/170, the authors write that the "interannual variability is the standard deviation of the detrended and de-seasonalized timeseries."**
**However, by doing so, they also include the contribution of the subannual variability.**

This is partly true, but we assume that most of the subannual signal is seasonal and remove the seasonal term; we have added a comment on this in the text.

**4) How did the authors account for the presence of sea ice in the Arctic Ocean? Does the satellite altimetry dataset provide sea-level observations in the regions covered by sea ice? If not, has this region been masked in the other datasets to ensure that the datasets return consistent results? In any case, I suggest that the include additional information on how the producers of the satellite altimetry data handle the presence of sea ice.**

We have included this information in Sections 2.1 and 2.2, thank you. Indeed, the data is flagged in the presence of sea ice in the C3S product and masked out. There is no data at very high latitude, because of the recurrence of sea ice.

**Results**
* * *
**General comment on this section**
* * *
**1) The text does not specify the period of the analysis. This should be clearly stated in the text.**

We have added this info at the beginning of section 3. Thanks

**Figure 1 suggests that the analysis spans the period between January 2003 and December 2019. However, it seems that all the datasets are available from April 2002 to December 2019. The authors should state why they did not perform the analysis over this longer period.**
Thanks. We excluded a few months (second half of 2002) over the 17 years to have a homogenous period covering full years, for all the analyses presented. We prefer not to include this information in the text, as this is probably not crucial for most readers.

**2) Section 3.3 examines the relationship between large-scale atmospheric/oceanic patterns and manometric sea-level variations in the three regions. However, it lacks a thorough description of the results. More details are needed to show that the statistical method provides results that are physically sound.**
We have added more information and interpretation of the results, and a few more notes in section 2.4. Note however that this fingerprinting technique is based on a statistical method (LASSO regression) that by construction minimizes collinearities.

**Section 3.1**
* * *
**1) L194/195: There is typo either in the main text or in Table 2. The main text states that, in the Arctic Ocean, GRACE returns a manometric sea-level trend of 2.45±0.44 mm/year, whereas GREP of 3.45±0.57 mm/year. However, Table 2 shows the opposite.**

**2) L201: There is typo either in the main text or in Table 2. The main text states that the interannual variability in the North Atlantic Ocean ranges between 6.6 to 8.6 mm. However, Table 2 shows values between 6.0 and 6.6 mm.**

**4) L205: There is typo either in the main text or in Table 2. The main text says that the interannual variability of manometric sea level in the Mediterranean Sea is more than 25 mm for all datasets. However, Table 2 shows that the interannual variability from GREP has an amplitude of 20 mm.**

Thanks a lot for carefully checking the consistency of the results. The previous three points are due to some typos due to the recomputation between the original and revised versions of the manuscript. By mistake, we revised and upgraded the table but not all the texts. Now we have adjusted the main text and rechecked that all values in the Table are correct.

**3) L203: The authors write that, as expected, the North Atlantic manometric sea-level variability resembles the global signal. This statement needs to be supported by one or more references.**
Thanks for pointing this out; it was our subjective expectation, but as we did not find any clear indication of this in past literature, we removed "as expected" from the sentence.

**5) L211/212: I suggest the authors provide an explanation of why GREP tends to underestimate the maxima in manometric sea level in the Mediterranean Sea in 2006, 2010, 2011, and 2018.**
Added: "likely due to the use of climatological discharge from rivers in the reanalyses, and the low resolution at Gibraltar strait affecting the representation of the Mediterranean inflow".

Section 3.2
* * *
1) L225: The authors write that "… SLB might capture the year-to-year variations better than the reanalyses". I do not think the authors can make this statement as the correlation between GRACE and SLB is not statistically different from the correlation between GRACE and GREP.

2) L228: I recommend that the authors rephrase the sentence in which they argue that they have greater confidence in GRACE and GREP than in SLB regarding the seasonal cycle in the Mediterranean Sea. This seems an important conclusion, but it is not very well expressed.

**3) L231-238: The authors state that the SLB approach returns poor results in the Arctic Ocean. This is an interesting conclusion. However, as the manuscript aims to compare the quality of the different datasets and approaches, it seems important to investigate this point further and try to understand whether the poor performance with the SLB approach results from problems in the sea-level anomaly or in the temperature and salinity profiles. For example, how does the sea-level anomaly from satellite altimetry compare to that provided by the reanalyses? Or how does the steric sea level from observations compare to that provided by the reanalysis?**

We thank the reviewer for this suggestion, and we compared, in terms of correlation, the reanalyses total and steric separately with the altimetry and steric dataset used within the SLB approach. What we found (partly expected due to the assimilation of altimetry data in reanalyses) is that the total sea level between the two is quite well correlated (e.g. 0.69 for the interannual signal) while the steric is weakly correlated (0.35), resulting in an overall small correlation as shown in the figure. We cannot add much to this in terms of figures, etc., but we added a sentence in section 3.2 to report this finding.

Section 3.3
* * *
**1) L254/L255: The authors write that: "AMO is known to modulate the sea-ice interannual variations and the Arctic amplification (Li et al., 2018; Fang et al., 2022), which are both important contributors to the sea level manometric fluctuations." The authors should provide evidence that sea-ice interannual variations and the Arctic Amplification significantly affect the manometric sea-level variability in the Arctic.**

Over the past few decades, the Arctic Ocean has been experiencing rapid warming, a phenomenon known as Arctic Amplification. Fast warming has been leading to a widespread shrinkage of the cryosphere, including sea ice, but also glaciers, and ice sheets. These rapid changes are affecting the atmospheric and ocean circulations in the Arctic, which in turn impact the climate variability at both regional and global scales. The increased melting of land ice and disturbances in atmospheric and ocean circulation influence the variability of manometric sea levels through the input of freshwater and its redistribution in and out of the Arctic basin. These significant and interconnected changes in the Arctic climate are caused by multiple factors, including human-induced greenhouse gas emissions, as well as internal climate variability such as the AMO.

We cannot detail all these arguments, but we added a reference (Previdi et al., 2021) that addresses these issues in detail.

Previdi, M., Smith, K. L., & Polvani, L. M. (2021). Arctic amplification of climate change: a review of underlying mechanisms. Environmental Research Letters, 16(9), 093003. doi:10.1088/1748-9326/ac1c29

**2) I recommend that the authors show the spatial patterns of the climate modes, and the timeseries of the respective climate indices. Journal restrictions might prevent the**

**authors from doing it. In this case, could the authors add this piece of information in the supplementary material?**

The time series and spatial patterns of climate modes are shown in Fig. 1, 3, and 4 in Pfeffer et al., 2022, and are not reported in the paper to avoid redundancy and for the sake of brevity as required by the journal policy; the link with the previous paper is made now more evident in section 2.4.

**3) I would also suggest that the authors better explain their results in section 3.3. As an example, why does the manometric sea level in the North Atlantic appears to be more related to the NPGO than to the NAO, the AO, and AMO? The authors provide little information on the reasons behind this relationship. They argue that: "(L260-261) While NPGO well explains variations in the eastern North Pacific Ocean (Di Lorenzo et al., 2008), its impact on the North Atlantic manometric sea level likely depends on the global barystatic signal and teleconnections (Iglesias et al., 2018)." Which teleconnections in Iglesias et al. can explain this relationship? Furthermore, do Iglesias et al., 2018, focus on the entire North Atlantic Ocean or only on the eastern North Atlantic Ocean?**

The NGPO index is defined as the second mode of sea surface height variability in the Northeast Pacific (180°W–110°W; 25°N–62°N). However, The nature of PDO and NPGO is changing at multidecadal time scales in a warming climate. The spatial structure of such modes is non-stationary in time, and the relationship between the physical, environmental, and biological variables and the climate indices evolves. The correlation between the PDO and NPGO has increased in the last decades. In recent decades, the NPGO showed increasing association with the first mode of climate variability rather than the second (Litzow et al., 2020). The NGPO may therefore be a relevant index to understand climate variability at the global scale, which influences manometric sea level changes in the Atlantic. It may be noted that Pfeffer et al., (2022), also found a widespread correlation of the NGPO with the water mass redistribution observed with the GRACE and GRACE-FO satellites, although this association was found to be more robust on land than on the ocean. To avoid confusion and subjective interpretations, we now refer only to the published work showing the wide impact of the index on the sea level and have rephrased the sentence.

**4) The authors try to identify the large-scale atmospheric and oceanic patterns that are responsible for manometric sea-level variations in the North Atlantic. However, this region is wide as it extends from 0° to 67°N. So, different areas of the North Atlantic might be affected by different large-scale patterns. I suggest that the authors focus on different sub-regions of the North Atlantic. For example, they could consider the tropical North Atlantic, the mid-latitude western North Atlantic, and the mid-latitude eastern North Atlantic separately.**

The purpose of this article is not to provide a detailed explanation of the factors affecting the variability of sea levels in the Atlantic Ocean and sub-basins. Instead, we aim to describe the changes in manometric sea level as accurately as possible using various datasets and evaluate the significance of the different signal contents in comparison to known modes of variability. Subdividing the NA in sub-basins would result in inconsistency with the analyses performed in Figures 1, and 2, and we prefer to keep the analysis as it is.

**5) The authors should also explain why the manometric sea-level variability in the Mediterranean Sea seems to be largely affected by the AO, but not by the NAO. In this respect, the authors consider the AO and the NAO as two distinct climate modes, but the AO and NAO indices might not be independent. On the contrary, they might be**

**highly correlated (e.g., Ambaum et al., 2001). How do the authors handle this statistical dependence? How much do the results change if either the NAO index or the AO index is excluded from their analysis?**

The LASSO will by construction select the indices in the multivariate regression that will minimize the cost function, expressed as the sum of least-squares residuals and absolute values on the coefficients of the regression. If two indices are largely correlated, the LASSO will pick the one minimizing the cost function. Shall we remove the AO from our analysis, NAO will be selected as a relevant index. The AO and NAO should not be considered as two independent modes, but rather as two relatively similar ways of describing the climate variability in the Arctic and North Atlantic regions. We added a sentence on that (in section 2.4 and while commenting on the results).

**6) The authors use the full manometric sea-level signal to study the relationship between manometric sea-level variations and climate modes. However, splitting the signal into interannual and subannual variability is needed as the two might be forced by different climate modes.**

The strength of a multivariate LASSO regression is to select all relevant modes of variability simultaneously. The method can select multiple indices, explaining either short-term or longer-term variations. The method however does not account for delays (e.g. out-of-phase signals) or seasonal dependence (e.g. summer or winter influences of a specific mode may differ) between climate indices and manometric sea level changes. Consequently, it is not very meaningful to filter the input signal given to the LASSO regression.

**7) The authors should explain how they derive the climate modes and the climate indices used in the manuscript. For example, which datasets and which techniques do they use?**

The time series and spatial patterns of climate modes are shown in Fig. 1, 3, and 4 in Pfeffer et al., 2022, to which we refer for the data sources, etc. (please see the answer and modification related to previous point 2).

**Table 1**

**--------**

**I would recommend the authors to rearrange the order to the products in Table 1. I would rearrange them in such a way that they follow the same order of appearance as in the "Data and Methods" section.**

**Else, the authors could remove Table 1. The piece of information in Table 1 could be included in the main text. I would favor this solution if it allowed for the inclusion of a new figure (e.g., a figure showing the spatial patterns of the climate modes used in Section 3.3 and of their respective indices).**

Thanks. We have changed the order of the products in the Table as requested; however, we prefer to keep the Table with full information, as this could be interesting for future follow-up studies and intercomparison (reading a few numbers within the text won't ease their use in future studies).

**Figure 2**

**---------**

**The colorbar does not help understand the heatmaps. I recommend the authors to test alternative colorbars. For an example, how would the viridis colorbar perform (https://matplotlib.org/stable/users/explain/colors/colormaps.html)? The authors could also try reducing the range of the colorbar as the lower correlations are higher than 0.2**

**or 0.3. Maybe, this would help make the figure clearer. Using a discrete colorbar and adding the actual number within each cell of the heatmaps could also help.**
**The authors could also reduce the size of the white spaces in between the subplots.**
Thanks for the suggestions, which have all been implemented in the revised figure.

**Figure 3**
**---------**
**The histograms contain vertical bars to show the standard errors of the regression coefficients. However, there is no vertical bar associated with the contribution of the NPGO to the manometric sea-level variability in the North Atlantic. Also, the limits of the y axes in Figure 3 should be modified because this same contribution is out of scale.**
Replotted as suggested; we also show the mean of the relative importance across the dataset, to provide a more "global" measure of impact.

---

## Author Response (AR3)

*SP-2023-28 Submitted on 02 Aug 2023*
*Reply to the reviewers. Third round of reviews*
*"Variability of manometric sea level from reanalyses and observation-based products over the Arctic and North Atlantic Oceans and the Mediterranean Sea" by Andrea Storto, et al.*

We thank once again the reviewer for the careful reading, the encouraging comments, and the suggestions to correct concepts and sentences in the manuscript. We report below a point-by-point answer to the Reviewer's comments (Reviewer in black, our reply in red).

**Reply to Anonymous Reviewer 2**

**Main comments**

- In relation to one of my previous comments (comment 3 on Section 2.4), the authors responded by stating that the majority of the subannual signal can presumably be attributed to seasonal variations. They mentioned that this assumption is discussed in the text, but I was unable to locate it. Could they guide me to the specific section? Alternatively, if this explanation is absent, they should incorporate it into the text.
Added in the first paragraph of section 2.4

- L15: the first letters in "global reanalysis ensemble product" should be capitalised.
Corrected

- L20: I think it would be better to substitute "is well explained" with "resembles". "Explained" might give the impression that the global barystatic sea level variations force sea level variations in the North Atlantic.
Changed to "highly correlated"

- L24: I do not know if "than in comparison with" is grammatically correct. In any case, it would sound wordy and redundant. The authors could remove "in comparison with".
Corrected

- L35-44: I recommend that the authors partly rewrite the first paragraph of the Introduction. The first line of the introduction explicitly refers to interannual variations. However, the first paragraph does also refer to longer timescales.
Also, the authors state that interannual sea level variations can be decomposed into a steric and a manometric component. However, this should be true at all timescales.
We have removed "at the interannual timescale".

- L35/36: The phrase "changes in density-driven variations in sea level" contain both the word "changes" and "variations". One of the two should be removed.
We have removed "changes"

- L143: Could the authors check whether "CGLORS" should be written "C-GLORS" instead?
Corrected

- L167: The "b" in "Bootstrapping" should not be capitalised.
Corrected

- L197: I am not sure if I can locate the part of the text where the authors explain the meaning of "mean uncertainty" and where they explain how they calculate it. Could they help me locate the specific line? Is it in the caption of Table 2? Or, if it is missing, could they include this explanation into the main text?

Added in the text, defined as the time-averaged uncertainty of each product, defined in turn as ensemble spread as specifiec in section 2.4

- L211: "Here, the manometric trend accounts for about 60-80% of the total sea level trend, depending on the specific product used." With "total sea-level trend", do the authors refer to the barystatic sea level trend?

We referred to the total (barystatic + steric) sea level provide by altimetry. Now explicated in the text to make it clear

- L213-L214: I think that a parenthesis is missing after "Table 3" and that one parenthesis should be deleted after "considered".

Corrected

- L220: The "s" in "strait" should be in capitalised.

Corrected

- L228: I think that "decomposing the full signal" should be followed by "into", not "in".

Corrected

- L247: maybe "possibly due" is better than "due".

Corrected

- L284-L285: The sentence "Other influential climate modes of variability are linked to the North Pacific variability, namely the PDO and NPGO" seems a little bit out of context. Could the authors rewrite this sentence for it to be better connected to the paragraph where it is located?

We have rewritten this last sentence to make the context more clear.

- L600: In the sentence "... whereas as the minimum correlation ...", "as" should be deleted.

Corrected

- Table 2: I recommend that the authors specify the units.

Added in the caption